# PROCEEDINGS A

statistics

power-law networks, Kolmogorov–Smirnov test, de Solla Price model

**Author for correspondence:**
E. C. Wit
e-mail: wite@usi.ch

# How rare are power-law networks really?

I. Artico[1], I. Smolyarenko[2], V. Vinciotti[2] and E. C. Wit[1]

[1]Università della Svizzera italiana, Lugano, Switzerland
[2]Brunel University London, Uxbridge, UK

IS, 0000-0003-0770-2366; ECW, 0000-0002-3671-9610

The putative scale-free nature of real-world networks has generated a lot of interest in the past 20 years: if networks from many different fields share a common structure, then perhaps this suggests some underlying 'network law'. Testing the degree distribution of networks for power-law tails has been a topic of considerable discussion. *Ad hoc* statistical methodology has been used both to discredit power-laws as well as to support them. This paper proposes a statistical testing procedure that considers the complex issues in testing degree distributions in networks that result from observing a finite network, having dependent degree sequences and suffering from insufficient power. We focus on testing whether the tail of the empirical degrees behaves like the tail of a de Solla Price model, a two-parameter power-law distribution. We modify the well-known Kolmogorov–Smirnov test to achieve even sensitivity along the tail, considering the dependence between the empirical degrees under the null distribution, while guaranteeing sufficient power of the test. We apply the method to many empirical degree distributions. Our results show that power-law network degree distributions are not rare, classifying almost 65% of the tested networks as having a power-law tail with at least 80% power.

## 1. Introduction

Networks play an important role in many fields, from epidemiology and ecology to engineering and sociology. They are a powerful way to represent and study the interaction structure of complex systems. An important measure of the network topology is the distribution of the number of connections per node: the *connectivity distribution* [1], also known as the *degree distribution*.

One contribution to a special feature 'A generation of network science' organized by Danica Vukadinovic-Greetham and Kristina Lerman.

Many empirical networks have been reported to exhibit *scale-free* behaviour based on the distribution of the connectivities of the network nodes [2,3]. Describing networks can be justified in two distinct ways: either phenomenologically based on network data or from first principles.

Power-law networks have been proposed as a 'universal' model, as they possess a number of important properties, such as the presence of hubs and large numbers of nodes with few connections [4] as well as a typical small-world behaviour [5]. The latter allows fast communication between nodes even for huge networks, given the small diameter characteristic of small-world networks. The definition of a power-law network varies across the literature, but one often cited definition is that its degree distribution $P$ satisfies $P(d) \propto d^{-\gamma}$, where $\gamma > 1$ [6]. Some versions make additional requirements, e.g. requiring that node degrees evolve via a preferential attachment mechanism [7], and specify, mathematically more correctly, that the power-law only should hold asymptotically in the upper tail of the degree distribution [2,8].

However, from a phenomenological point of view, observed networks are (almost) always finite, hence a power-law network is indistinguishable from a network with a sufficiently distant exponential cut-off of a power-law degree distribution. If our sole purpose is fitting an observed degree sequence, then a large class of models will do an equally good job for the types of networks we tend to encounter in practice. Nevertheless, ever since de Solla Price started to experiment with potential generative network models in the 1960s, it has become clear that a small number of substantively plausible and generative principles are capable of generating network structures that correspond to empirical networks. Particularly, various forms of preferential attachment rules have been shown to result in network structures whereby the degree sequences are generally described by ratios of gamma functions [9], i.e. power-laws. This putative universality of the power-law degree distributions sets it up as a natural paradigm for falsification [10], i.e. as a natural null hypothesis. It is from this epistemological point of view that we approach the question of power-law networks in this paper. On top of this, others have also argued that it is practically important to know whether networks are power-law, as such networks are, for example, more susceptible to epidemics and other viral events [11].

A long-standing issue in network science is how prevalent the power-law property is in empirical networks. A spate of early analyses, often using fairly crude methodology, resulted in a widespread acceptance of the belief that power-law degree distributions, viewed as a proxy for a network being scale-free, are quite ubiquitous [12–15]. This coincided with intensive theoretical efforts to explain the putative universality of power-law degree distributions. More recently, more sophisticated statistical techniques have cast doubt on the extent of the scale-free universality. Starting with the work of Khanin & Wit [16], biological networks were shown to fit better with a truncated power-law model, i.e. a power-law regime followed by a sharp drop-off, $P(d) \propto d^{-\gamma} e^{-d/k_c}$. The authors found that the number of connections in biological networks significantly differs from the power-law distribution and that these networks are not scale-free. Another critique was levied in a recent paper by Broido & Clauset [17], who use a likelihood ratio test within a nested testing procedure, suggesting that the evidence for power-law distribution is often weak. A drawback of these critiques is the emphasis on identifying 'pure' power-law tails as this leads to two conflicting requirements: a cut-off far into the distribution tail to ensure, in some sense, sufficient closeness to the asymptotic power-law, and the availability of a sufficiently large number of data points for meaningful statistical testing. The same issue has recently been highlighted by Voitalov *et al.* [8], who devise consistent estimation procedures for the exponent $\gamma$ taking into account the asymptotic nature of power-laws, but who reject the possibility of a formal testing procedure.

Even though a number of studies have considered testing for power-law degree distributions in empirical networks, the final verdict is still open. This current paper takes a complementary view to Voitalov *et al.* [8]: we make stronger parametric assumptions about the asymptotic form of the tail of the degree distribution, avoiding the impossibility arguments [8, Section V], in order to get a lower-bound on the fraction of empirical networks that exhibit power-law behaviour. This parametric assumption consists of assuming that the tail of the degrees comes from a de Solla Price network process, a two-parameter preferential attachment model. This does not mean that

a de Solla Price is a sensible model for real-world networks, but being a subset of the power-law distributions, not being able to reject with sufficient power a de Solla Price model would mean that we have positive evidence for a power-law tail.

In §2, we present the landscape of the main methodological issues encountered in testing degree distributions in empirical networks. In §3, we present the proposed testing framework. We present (i) a specific parametric asymptotic power-law model that will be used to test the goodness-of-fit of the empirical degree distribution, (ii) a modification of the classic Kolmogorov–Smirnov (KS) statistic to deal with dependent degree samples as well as heterogeneous variances and (iii) a way to calculate the power of the test-statistic. In §4, we apply the testing framework to 4482 empirical networks. Our aim is to decide whether in a large body of networks the power-law property holds or should be seen as too simplistic. In §5, we present our conclusions.

## 2. Issues in testing empirical degree distributions

In this section, we present an overview of the main issues encountered in testing whether empirical degree distributions are power-law. In particular, (i) we will introduce the exact asymptotic definition of a power-law degree distribution and relate this to the problem of observing only finite networks; (ii) we explain how the dependency of a single empirical degree sample affects the distribution of a KS test statistic and (iii) we show how asymptotic tests must balance the delicate equilibrium between power of the test and the asymptotic power-law property. The issues introduced in this section will be resolved in §3.

### (a) What is a degree distribution?

A simple random graph on vertex set $V = \{1, \ldots, N\}$ is defined by its graph distribution $H : E \rightarrow [0, 1]$, which associates with any graph $G$ a probability $H(G)$. For directed graphs with possible self-loops $E = \{0, 1\}^{N \times N}$, whereas for directed graphs without self-loops or undirected graphs $E$ is a strict subset of $\{0, 1\}^{N \times N}$. For any vertex $i$ in the graph $G$, we define its degree $d_G(i)$ as the number of edges in $G$ that involve vertex $i$. In the case of directed networks, one could focus on the in-degree or out-degree instead, but this will not change the exposition below. Given a particular degree definition, we define the *marginal degree distribution* $P(\cdot|i) : \{0, \ldots, N\} \rightarrow [0, 1]$ *for vertex i* as the probability over all graphs $G$ for which vertex $i$ has a particular degree,

$$P(d|i) = \sum_{d_G(i)=d} H(G).$$

Two important points to note are that the measures $P(\cdot|i)$ and $P(\cdot|j)$ for $i \neq j$ are generally *dependent* and *not identical*. Only if the measure $H$ is exchangeable, then the marginal degrees are identically distributed. Only in very special cases, such as for certain types of Erdős–Rènyi graphs, these marginal degrees are both independent and identically distributed.

The *average degree distribution* $P : \{0, \ldots, N\} \rightarrow [0, 1]$ is defined as the marginal degree distribution of a randomly selected vertex,

$$P(d) = \frac{1}{N} \sum_{i=1}^{N} P(d|i).$$

We will refer to this distribution simply as *the degree distribution*. In fact, it is this distribution that one commonly considers in practice, for example, by plotting the histogram of degrees of all the vertices in a particular graph.

For graphs with infinitely countable vertex sets, the same definition for the marginal degree distribution can be given, whereas the (average) degree distribution is defined as a limit,

$$P_{\inf}(d) = \lim_{N \to \infty} \frac{1}{N} \sum_{i=1}^{N} P(d|i).$$

For the Barabási–Albert preferential attachment model, it can be shown that $P_{\inf}(d) = 4/((d+1)(d+2)(d+3))$ for the in-degree $d \in \mathbb{N}_0$ [7].

We define *power-law degree distributions* as those degree distributions for infinite graphs that possess a particular asymptotic property in their tail. In particular, an infinite graph degree distribution $P_{\inf}$ is considered *power-law* if there exists a $\gamma > 1$ such that

$$\lim_{d \to \infty} d^{\gamma} P_{\inf}(d) = c, \tag{2.1}$$

where $c > 0$ is an arbitrary positive constant, e.g. for the Barabási–Albert preferential attachment model $\lim_{d \to \infty} d^3 P_{\inf}(d) = 4$. This definition of a power-law is more restrictive than the regular variation definition in Voitalov *et al.* [8], but this is sufficient for our purposes.

## (b) Finitely observed network

As any empirically sampled network is finite, in what sense can this finite network be related to the power-law? Since a vertex in a simple graph without self-loops cannot have more connections than the total number of vertices excluding itself, the degree distribution has a support that is bounded above by $N - 1$. This means that it is impossible to detect scale free networks, whose power-law regime 'starts' at $O(N)$. *Every finite network degree distribution could potentially behave like a power-law on the unseen degrees*. That is why, strictly speaking, talking about power-law degree distributions for finite networks is meaningless.

However, if the finite network is, in a certain sense, a 'random sample' from an infinite network, then under certain conditions it might be possible to relate the finite sample degree distribution to the infinite population distribution. Sampling subnetworks is more complicated than sampling ordinary populations, as specific choices have to be made: whether to sample primarily vertices or edges and how to sequence the sampling. Most state-of-the-art network sampling schemes, i.e. link tracing, star, snowball, induced and incident sampling, have drawbacks that lead to certain biases in the estimation of the degree frequencies [18, ch. 5.6]. We will show how certain generative sampling assumptions will allow us to sample finite networks that asymptotically form a subclass of the power-law degree distribution networks.

## (c) Dependent versus independent degree samples

Essentially all existing work on empirical degree distributions (e.g. [8,17,19–22]) treats the observed degree sequence of an empirical network as an *independent* random sample. However, depending on the underlying random graph distribution, observing a degree for a particular node may well be positively or negatively correlated to the degree of another node. A sample of degrees coming from a single realization of a network should, therefore, be considered as a dependent sample. The impact of this dependence on test-statistics that involve the empirical degree distribution has not been studied in any detail until now.

Smolyarenko [23] shows that tests based on the empirical degree distribution can have markedly different behaviour from what would be expected under independence. In particular, the scaled empirical cumulative distribution function for degree distributions in standard synthetic networks does not converge to a Brownian bridge [24]—see appendix A for details. We will show that under certain network distributions the variance of the empirical degree distribution is lower than expected under independence, invalidating traditional KS tests.

## (d) Power of goodness-of-fit test

As we want to test the null hypothesis that an empirical degree distribution comes from a power-law network, it is important to be able to control the power of the goodness-of-fit test. Regardless of the test choice, not rejecting '$H_0$ : *network is power-law*' is not necessarily proof of the validity of $H_0$ without additional control of the power of the test. Power of a test controls the probability of rejecting $H_0$ when it is false. Although one clearly desires a high level of power in order

to correctly detect power-law networks, this does not come for free: it involves determining the level and type of departure of power-law that is practically insignificant. We will make recommendations on how to set sensible values for this allowable deviation.

Furthermore, since a power-law is a tail property, the test statistic will focus on the tail of the degree distribution. This leads to two, possibly conflicting requirements, since the further along in the tail of the degree distribution we check, (i) the more likely our parametric power-law distribution is able to fit a power-law tail if it is present, but (ii) the less power the goodness-of-fit has to detect it. We have to find a balance between, on the one hand, testing the tail and, on the other hand, having sufficient tail observations to guarantee a certain power of the test.

## 3. Testing framework

In this section, we present an integrated testing framework that addresses the issues that were described in §2. Our aim is to describe a comprehensive procedure that based on a non i.i.d. degree sequence from a finite network is able to test the null hypothesis

$$H_0 : \text{The degree distribution } P_{\text{inf}} \text{ is power-law,}$$

where the finite network is assumed to be a particular type of sample of $P_{\text{inf}}$ as described in §3a. Then in §3b, we operationalize the concept of a power-law degree distribution by means of a flexible, generative family of degree distributions. In §3c, we introduce a modified KS test statistic that deals with all the difficulties we identified above and in §3d we show how we can control the power of this test.

### (a) Sampling finite networks

Empirical finite networks can occur in many different ways [25]. It could be that the vertex set is fixed and the edges are drawn from some distribution. These networks are not of interest to us in this manuscript. Clearly, such non-growing networks have no relationship with any underlying, infinite network distribution that might or might not exhibit power-law behaviour. Instead, in this manuscript, we assume that $P_{\text{inf}}$ is the resulting degree distribution from a generative and additive network sampling scheme that at each moment can be stopped to obtain a finite network.

For example, the Barabási–Albert preferential attachment model is a generative network sampling scheme that at each step adds a vertex to the network that it connects to one of the other vertices already in the network with a probability proportional their degrees. This procedure can be stopped for any finite size $N$ network, leading to a degree distribution $P_N(d)$. Whereas the finite Barabási–Albert preferential attachment model converges to a network with a power-law degree distribution, other iterative sampling schemes might not.

### (b) A finite de Solla Price power-law

As the power-law property is a mere asymptotic characteristic of a network, the class of power-law networks is vast. On purpose, we will restrict ourselves in this manuscript to a subfamily of power-law networks. As our main assumption in §3a is that the finite network is in a generative way associated with the infinite network measure, we will focus on a generative class of power-law distributions, namely preferential attachment models. These models iteratively extend the network, both in terms of vertices and edges, in such a way that networks of any particular size can be achieved.

Krapivsky & Redner [9] describe a rich class of network models constructed by means of a general generative preferential attachment procedure with arbitrary connection kernels. They show that these kinds of models result in degree distributions that can be described by ratios of gamma functions. Ratios of gamma functions are the discrete analogues of power-laws. Using finite gamma ratios as a model for power-law degree distributions has the crucial advantage of treating some of the 'midsection' of the degree distribution as signal rather than noise. Broido &

Clauset [17], Khanin & Wit [16] and others have been unnecessarily restrictive trying to find pure power-laws rather than to accept that some aspects of curving in log–log plots are informative, starving typical power-law tests of data.

We focus on a particular two-parameter gamma ratio model, known as de Solla Price model introduced in 1965 for modelling growing citation networks [26]. In the context of a growing network, $m$ is the number of new edges added to the network at each iteration of the growing algorithm and $d + w$ is proportional to the preferential attachment probability for the vertices with $d$ incoming links. Van der Hofstad [27] and Newman & Girvan [28] show that the infinite degree distribution is given by

$$P^{sp}_{\inf}(d; w, m) = c_{m,w} \frac{\Gamma(d + w)}{\Gamma(d + 2 + w + w/m)},$$

where $0 < w < \infty$, $m \in \mathbb{N}$ and the normalizing constant $c_{m,w} = (1 + (w/m))(\Gamma(1 + w + w/m))/\Gamma(w)$. The model is a generalization of the Barabási–Albert model, which is the special case when $m = w = 1$ and $d \in \mathbb{N}_0$ is the in-degree. Combinations of the parameters $(w, m)$ allow for more flexibility and the model is therefore better able to capture empirical distributions at lower degrees. As $d \to \infty$ the model shows a power-law behaviour proportional to $d^{-\gamma}$, i.e.

$$P^{sp}_{\inf}(d; w, m) = c_{m,w} d^{-\gamma}(1 + O(1/d)),$$

where $\gamma = 2 + w/m$ [27].

The finite de Solla Price degree distribution of size $N$ is denoted as $P^{sp}_N(\cdot; w, m)$. We will use $F^{sp}_N(d; w, m) = \sum_{i=0}^{d} P^{sp}_N(i; w, m)$ as notation for the cumulative distribution function of the finite de Solla Price model. Although the de Solla Price model is flexible and can fit a wide range of empirical power-law degree distributions, the model is still not flexible enough for our purposes. In order to address this issue, we define a model that behaves as de Solla Price on the degrees above a specified cut-off $c$ and is free to take any other shape for the degrees below, in particular

$$P^{sp}_{c,N}(d; w, m) = \begin{cases} p_k & d = 0, \dots, c-1 \\ P^{sp}_N(d; w, m) & d = c, \dots, N-1 \end{cases}$$

with its associated cumulative degree distribution function $F^{sp}_{c,N}(\cdot; w, m)$. Barabási [19] suggested that power-law networks often have such low degree deviations, which should be ignored. We refer to this network model as the extended de Solla Price network model, which is generated by arbitrarily rewiring of edges between low-degree vertices.

## (c) A weighted Kolmogorov–Smirnov testing procedure

Given the de Solla Price subclass of power-law networks our aim is to test the more stringent null hypothesis

$H_0$ : *The network is drawn from an extended de Solla Price network model,*

based on a single finite empirical network sample. The idea is that the number of non-rejected tests, each with sufficient power, will give us an idea of the lower bound on the ubiquity of empirical power-law networks.

### (i) Traditional Kolmogorov–Smirnov test statistic

Traditionally the KS test statistic is one of the common statistics used to test for the goodness-of-fit of a particular presumed distribution of the data. It is defined as the largest distance between the

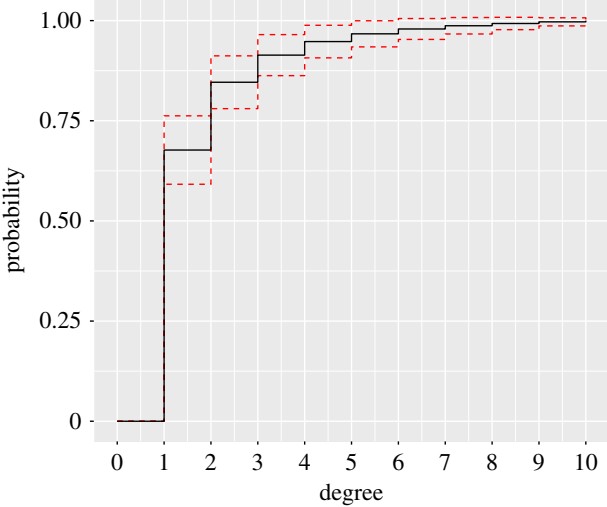

**Figure 1.** An example of a de Solla Price cumulative degree distribution; dashed lines indicate the standard deviation of the empirical degree distribution considering a network of size 30. (Online version in colour.)

empirical cdf and the hypothesized one,

$$D_{KS} = \sqrt{N} \sup_{d \geq 0} \left| \hat{F}_N(d) - F_{c,N}^{sp}(d; w, m) \right|, \tag{3.1}$$

where $d$ stands for the degree, $F_{c,N}^{sp}$ and $\hat{F}_N$ are, respectively, the true (under $H_0$) and the empirically observed degree distributions, $N$ is the overall number of observations, i.e. the number of vertices in the empirical network. The empirical degree distribution is defined as $\hat{F}_N(d) = (1/N) \sum_{v \in V} \mathbb{1}_{\{d_v \leq d\}}$ where $d_v$ is the observed degree of vertex $v$. Under the independent sampling assumption, the $D_{KS}$ statistic converges in distribution to the Kolmogorov limit distribution [29]. The convergence of $D_{KS}$ to the Kolmogorov limit distribution is based on the assumption of continuous data and independent observations, both of which are violated in the case of an empirical degree distribution from a single network. As shown by Smolyarenko [23], the KS test statistic for empirical degree distributions in evolving networks does not converge to the usual Kolmogorov limit distribution.

## (ii) Variance of the empirical degree distribution

As pointed out by Anderson & Darling [30], the KS statistic does not achieve uniform sensitivity over all quantiles. Under the independent sampling assumption, for a fixed degree $d$, we have that

$$N\hat{F}_N(d) \sim \text{Bin}(N, F_{c,N}^{sp}(d; w, m)), \tag{3.2}$$

with variance $NF_{c,N}^{sp}(d; w, m)(1 - F_{c,N}^{sp}(d; w, m))$. Although the independence is a rather unrealistic assumption, it can give an insight into the variance behaviour in empirical cumulative degree distributions. In particular, $\hat{F}_N(d)$ achieves its highest variance at $d = F_c^{sp-1}(0.5)$ and decreases to zero in the tails—in particular, in the right tail in case of a degree distribution. The distances $|\hat{F}_N(d) - F_{c,N}^{sp}(d; w, m)|$ are not identically distributed over $d$ and, more importantly, the decrease of the variance leads to a decrease of the sensitivity in the tail of the degree distribution. In typical network scenarios, this means that the KS statistic is mainly influenced by low degrees, whereas one mainly wants to detect deviations for high degrees. For example, figure 1 shows an empirical degree distribution, whose first degree $d = 1$ takes 66% of the overall probability and therefore is the main contributor to the KS statistic.

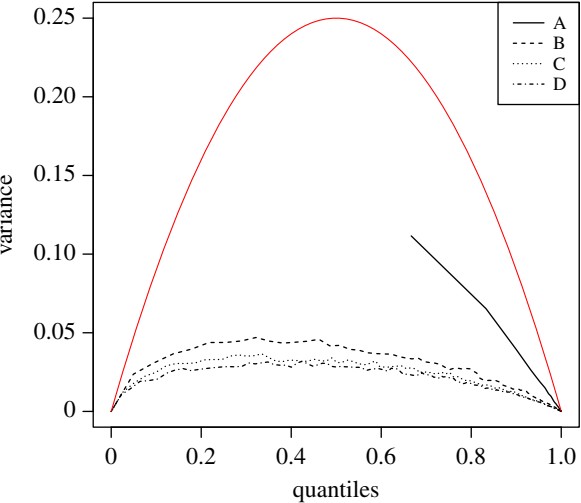

**Figure 2.** Brownian Bridge's empirical variance with A:$(w = 1, m = 1)$, B:$(w = 134, m = 23)$, C:$(w = 267, m = 44)$, D:$(w = 400, m = 51)$. Top line (red in online version) is the variance under independent degree sampling (see appendix A). Line A is complete, but starts from the first rescaled degree $F^{sp}(0; 1, 1) = 0.66$. (Online version in colour.)

Our aim is to modify the KS statistic in such a way that it achieves even sensitivity across the empirical degrees. Beyond the uneven variance addressed by the Darling–Anderson modification [30] described above, there are three additional considerations that affect the behaviour of the empirical degree distribution. In particular, we show how (i) the estimation of the parameters $(w, m)$ and (ii) the dependence among the empirical degrees lead to a reduction of variance, whereas (iii) the randomness of the observed degrees inflates the variance as compared to the independently sampled binomial case in (3.2) that we consider as our baseline.

(i) *Variance reduction due to parameters estimation.* In order to be able to calculate the KS statistic, one needs to estimate the parameters of the de Solla Price model. We use maximum likelihood to estimate its parameters. In particular, given a fixed value for $c$, we estimate the lower degree probabilities by their empirical counterparts. As the empirical distribution function and the MLE of the flexible de Solla Price coincide for low degrees, we have $|\hat{F}_N(d) - F^{sp}_{c,N}(d; w, m)| = 0$ for $d < c$. In general, estimation of the parameters reduces the variance of the KS statistic [31].

(ii) *Variance reduction due to dependent observed degrees.* As described in §2, the empirical degree distribution is a dependent sample of degrees. We will show that this affects the distribution of KS statistic $D_{KS}$. Chicheportiche & Bouchaud [32] show that the behaviour of the KS statistic, can be studied by analysing the random function $Y(u) = \sqrt{N}(\hat{F}(F^{sp-1}_{c,N}(u)) - u)$, $u \in [0, 1]$ is the $u$th theoretical quantile, since $D_{KS} = \sup_u y(u)$. If $\hat{F}$ was estimated by independent observations, then (3.2) would imply that $V(Y(u)) = u(1 - u)$. This is shown as the top line (red in online version) in figure 2.

Although the correlations between the empirical degrees are only of order $1/N$, the fact that there are $\binom{N}{2}$ of them, has a dramatic impact on the overall variance of $Y(u)$ and therefore on the KS statistic $D_{KS}$ [23]. We simulated from the de Solla Price preferential attachment model, using different values of $w$, the preferential attachment probability of the nodes with no incoming links, and $m$, the number of new links that each new node makes with the remaining nodes at each iteration of the growing process. Figure 2 shows that in all the scenarios the observed variance of $Y(u)$ and therefore $D_{KS}$, was lower than expected under independence. The negative correlations between the empirical degrees results in a significantly lower variance. This clearly casts doubt on a large scale of methodologies and past results which were based on the independence assumption (e.g. [6,17]).

(iii) *Variance inflation due to randomly observed degrees.* The baseline case, as described in (3.2), holds only for *fixed* degrees $d$ under the independent sampling assumption. However, the supremum taken in (3.1) will occur at an observed, i.e. *random* degree. As Goldman & Kaplan [33] showed for continuous distributions, the empirical degree $\hat{F}_N(d_{(i)})$ has beta distribution, i.e. $\hat{F}_N(d_{(i)}) \sim \beta(i, N + 1 - i)$, which holds approximately for high degrees due to the near-continuous behaviour of $\hat{F}_N$ in the degree tail for large networks. This results in a higher variance of the KS statistic than the binomial one. Clearly, this is true under the independent sampling assumption. For empirical degree distributions, it is challenging to quantify the overall variance inflation due to the degree randomness since we also have to consider the possible variance deflation due to the previous points.

### (iii) A modified Kolmogorov–Smirnov test statistic

Here, we will describe a test statistic that resolves the uneven variance, the reduced variance and the inflated variance that the KS statistic experiences for empirical degree distributions. As it is impossible to calculate analytically the effect of the various complicating factors, we resort to bootstrapping in order to define a uniformly sensitive, KS-like test statistic for testing the null hypothesis of a de Solla Price power-law degree distribution. This is possible because the de Solla Price is a generative network model, which can be sampled efficiently.

In particular, we consider an empirical network, for which we want to test whether it might have appeared from a finite de Solla Price network, $F_{c,N}(\cdot; w, m)$. We will assume that the cut-off $c$ is given—its value involves power considerations, described in §3d.

First, we estimate the parameters of the model $(w, m)$ from the data. A number of methods are proposed in the literature for power-law estimation, such as the Hill estimator for the tail coefficient of Wang & Resnick [34] and the maximum-likelihood approach on the network evolution data of Gao & van der Vaart [35], whereas a comparison between different estimators is provided in Clauset *et al.* [6]. In our framework, we estimate the unknown parameters $(w, m)$ by numerically maximizing the pseudolikelihood

$$L(d; w, m) = \prod_{i=1}^{N} P_{c,N}^{sp}(d_i; w, m),$$

via an iterative algorithm [36]. Crowder [37] showed that these estimates are consistent. For fixed discrete values of $m$, we maximize the likelihood according to $w$. We repeat the maximization procedure for a reasonable range of $m$ values. Finally, we select the $(m, w)$ values with the highest likelihood. This procedure is known as *profile* pseudolikelihood maximization. Further generalizations might be possible by specifying a random $m$ parameter [38] that can be sampled among the most likely values.

Then we define the test statistic $T$ as

$$T = \sqrt{N} \max_{v: d_v \geq c} \left[ \frac{\left| \hat{F}_N(d_v) - F_{c,N}^{sp}(d_v; \hat{w}, \hat{m}) \right|}{\sqrt{\hat{z}(d_v, \hat{w}, \hat{m})}}, \lim_{a \to d_v^-} \frac{\left| \hat{F}_N(a) - F_{c,N}^{sp}(a; \hat{w}, \hat{m}) \right|}{\sqrt{\hat{z}(a, \hat{w}, \hat{m})}} \right], \tag{3.3}$$

where $\{d_v\}$ are the observed degrees on the vertex set $V$ of size $N$ and $\hat{z}$ are the Monte Carlo estimated variances of the empirical degree distribution at the observed degrees for simulated de Solla Price networks with parameters $(\hat{w}, \hat{m})$. The distribution of the test statistic $T$ under the null hypothesis is obtained via a parametric bootstrap [39]. The parametric bootstrap consists of sampling degree distributions from the null hypothesis, i.e. a de Solla Price network generating process. The unknown parameters $(w, m)$ are substituted with the maximum-likelihood estimates, meaning sampling from the most likely de Solla Price distribution according to the observed data. We calculate the test statistics $T$ on each of them and obtain $T_1, \ldots, T_B$ bootstrap realizations of the test statistics distribution under $H_0$. We reject the hypothesis that the data come from a de Solla Price network if the test statistic $T^{obs}$ calculated on the observed network is greater than the 95% empirical percentile of the bootstrap distribution.

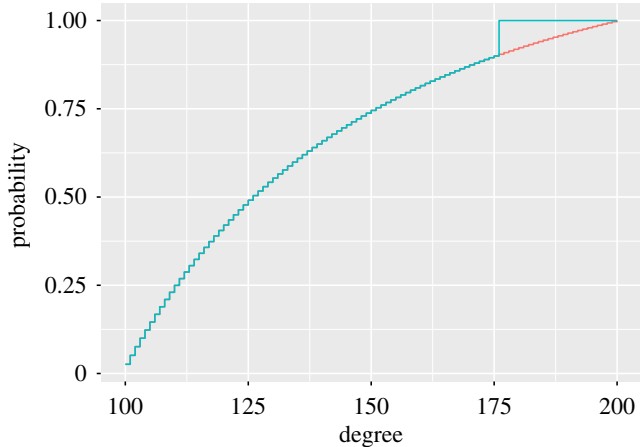

**Figure 3.** Under $H_0$ degree distribution is the 'continuous' (red in online version) conditional de Solla Price power-law, whereas under $H_1$ the degree distribution is taken to be the 'discontinuous' (blue in online version) function with $h_c = 0.1$ and $c = 100$. (Online version in colour.)

## (d) Cut-off choice via power analysis

This section selects the cut-off point $c$, by considering how many observations are left in the tail of the empirical degree distribution in order to guarantee the required power level. Although power is loosely defined as P(reject $H_0$ | $H_1$ is true), for continuous alternatives one needs to select a required minimum detectable effect size [40], which we define as the maximal distance $h$ between the true distribution and the null distribution $F_{c,N}^{\text{sp}}(\cdot; \hat{w}, \hat{m})$.

Power-law distributions decrease to zero slower than any other candidate distribution. Thus we choose an alternative distribution that decreases faster in the tail. Among all the possible degree distributions with at least $h$ maximum distance, the one that minimizes the power is the degree distribution that is exactly the same as the null $F_{c,N}^{\text{sp}}(\cdot; \hat{w}, \hat{m})$, but with a step of size $h$ placed in the end of the tail, as shown in figure 3. For values of $h$ that are sufficiently small, the distribution can be even closer to the power-law than the log-normal degree distribution. This assures that, once we fix the power for this type of function, all the other degree distributions that are $h$ removed from the de Solla Price power-law will have greater power, i.e. will be detected more easily.

In the practical analyses, in §4, we take a very stringent choice for the cutoff. In particular, we decided to calibrate $h = h_c(1 - F_N^{sp}(c; w, m))$ with $h_c \in [0.01, 0.1]$. This means that we aim to be able to detect degree distributions that have tail behaviour that decays faster on roughly the last 0.001 of the degree distribution. We choose a power of 80%, which means that if the true distribution differs from a power-law by only $h$ or more in the tail, then 80% of the time our method will detect it and reject the null hypothesis.

The power calculations are done straightforwardly by simulating $B = 200$ degree samples from the de Solla Price model, with maximum-likelihood estimated parameters. Then each sample is censored in correspondence with the degree in which the step occurs, obtaining samples from $H_1$. The statistical test is applied to each of them and the power is finally computed as the rate of rejected tests.

## (e) Overview of testing procedure

This section provides an overview of all the elements that go into testing whether a given empirical network comes from some de Solla Price power-law model. In five steps, the proposed

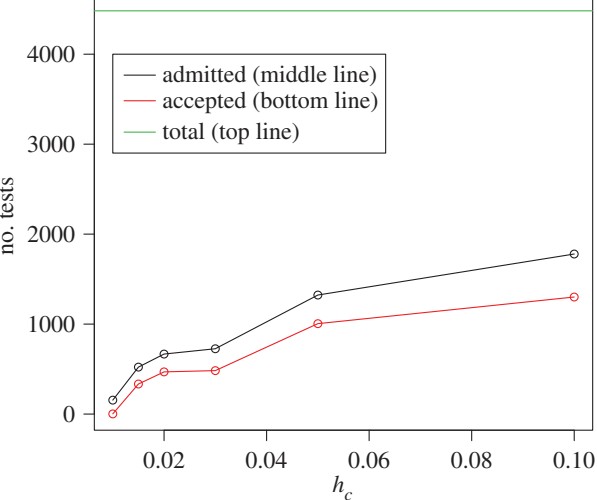

**Figure 4.** The top line (green in online version) shows the overall number of 4482 degree distributions that are possible to test. The middle line (black in online version) shows the number of admissible tests that have power greater than 80% with respect of tail sensitivity $h_c$. The bottom line (red in online version) illustrates the number of test for which the de Solla Price power-law seems to be a sensible model. (Online version in colour.)

testing procedure takes into account the power, degree dependency, cutoff and an even sensitivity over the tail of the test statistic.

(i) Step 1: calculate maximum-likelihood estimate on the original sample.

 (a) Fix the cut-off $c$ (for different values of $c$).

 (b) Given an observed degree sequence of size $N$, estimate $\hat{F}_N(\cdot)$ and $F^{sp}_{c,N}(\cdot; \hat{w}, \hat{m})$, where $\hat{w}$ and $\hat{m}$ are the maximum-likelihood estimates of the de Solla Price model.

(ii) Step 2: test distribution and variance computation.

 (a) Select number of bootstrap samples $B = 200$.

 (b) Generate $d_1, \ldots, d_B \sim P^{sp}_N(\cdot; \hat{w}, \hat{m})$ degree sequences with the de Solla Price preferential attachment algorithm up to a network with $N$ nodes.

 (c) Estimate the empirical degree distribution $\hat{F}^b_N(d)$ and the best fitting de Solla Price model $F^{sp}_{c,N}(d; \hat{w}^b, \hat{m}^b)$ for each of the bootstrap samples $b = 1, \ldots, B$.

 (d) Estimate the bootstrap variance $\hat{z}(\cdot; \hat{w}, \hat{m})$ of the difference $|\hat{F}_N(d) - F^{sp}_{c,N}(d; \hat{w}, \hat{m})|$.

 (e) For each bootstrap replication, calculate the test statistic, $T^1, \ldots, T^B$ using equation (3.3).

(iii) Step 3: test distribution under the alternative hypothesis with tail jump $h_c$ as shown in figure 3:

 (a) Fix the step size $h_c \in [0.01, 0.1]$.

 (b) Truncate $d_1, \ldots, d_B$ according to $h_c$, obtaining $d^{H_1}_1, \ldots, d^{H_1}_B$.

 (c) Estimate $\hat{F}^b_N(\cdot)$ and $F^{sp}_{c,N}(\cdot; \hat{w}^b_{H_1}, \hat{m}^b_{H_1})$ on the basis of $d^{H_1}_b$, with $b = 1, \ldots, B$.

 (d) Calculate the test statistics, $T^1_{H_1}, \ldots, T^B_{H_1}$.

(iv) Step 4: calculate $p$-value and power

 (a) Calculate the test statistic on the original data $T^{obs}$.

 (b) Calculate the $p$-value as the rate of bootstrap statistics that exceed the original statistic $p$-value$= \frac{\sum_{b=1}^B \mathbb{1}(T^b > T)}{B}$, where $\mathbb{1}(\cdot)$ is the indicator function.

 (c) Obtain $T_{0.95}$ as the 95% quantile of the bootstrap distribution.

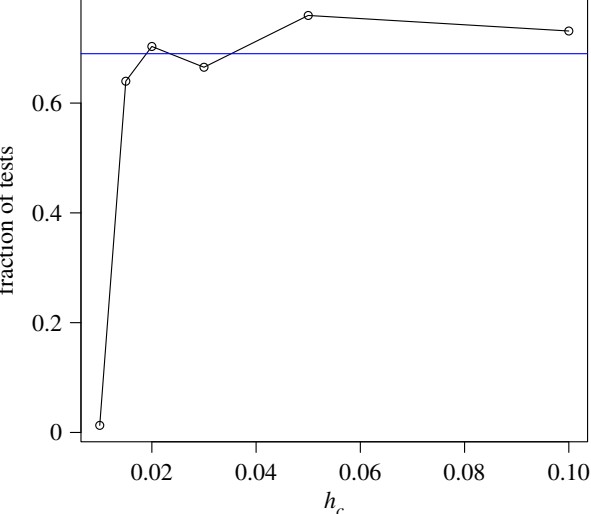

**Figure 5.** Fraction of accepted tests, i.e. rate of detected power-law networks, for $h_c = [0.01, 0.015, 0.02, 0.03, 0.05, 0.1]$. Note that the rate is stable for $h_c > 0.01$. This suggests that roughly two thirds of all considered real-world networks seem to exhibit power-law tail behaviour. (Online version in colour.)

(d) Calculate the power as the rate of $H_1$ statistics that are rejected by the test power= $(\sum_{b=1}^{B} \mathbb{1}(T_{H_1}^b > T_{0.95}))/B$.

(e) Select the largest $c$ for which the power is at least 80%.

## 4. Testing 4482 network for power-law degree distributions

We applied our testing framework to the datasets reported in Broido & Clauset [17], which consists of a large corpus of nearly 1000 network datasets drawn from social, biological, technological and informational sources. From these networks, the authors derived 4482 observed degree sequences. The corpus of real-world networks includes both simple graphs and networks with various combinations of directed, weighted, bipartite, multigraph, temporal and multiplex networks.

Similar to the authors in the original paper we are interested in testing whether the networks exhibit power-law degree distributions. For each degree distribution, we applied our testing framework for several values of the tail sensitivity $h_c = [0.01, 0.015, 0.02, 0.03, 0.05, 0.1]$, fixing a cut-off $c$ at degree 10. For lower values of cutoff, the test tends to reject most of the networks as de Solla Price, because of the other regimes present in the lower degrees that are irrelevant for power-law tail behaviour.

By fixing $c$ and $h_c$, it may occur that various networks do not achieve the required power of 80%. Those networks are excluded. Figure 4 shows the absolute number of degree distributions that are admissible to being tested, i.e. with power higher than 80%, as well as the absolute number of accepted tests, i.e. tests for which the power-law null distribution could not be rejected.

Figure 5 shows the $H_0$ acceptance rate over different $h_c$ values, as the rate of the non-rejected power-laws over the total number of tested networks. The lower $h_c$, the lower is the number of admissible networks to be tested. Nevertheless, the rate of networks for which the de Solla Price power-law cannot be rejected is almost constant for $h_c > 0.01$. Using the common elbow rule [41], a common practice among engineers, we select a very strong tail sensitivity $h_c = 0.015$ for which 64% of the tested networks exhibit power-law behaviour. For each of the non-rejected networks, we calculate the power-law exponent, $\hat{\gamma} = 2 + \hat{w}/\hat{m}$, with estimated parameters shown in figure 6. We find that for the more restrictive tests ($h_c = 0.015$), all the exponents are between 2 and 3,

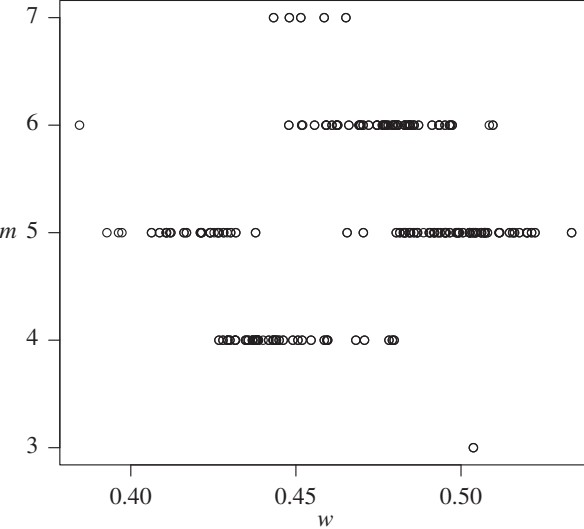

**Figure 6.** Estimates of $(w, m)$ for accepted tests at $h_c = 0.015$.

whereas for the most liberal tests ($h_c = 0.1$), 99.1% of all exponents are associated with what is normally called scale-free power-laws. As this acceptance rate stays constant for increasing values of $h_c$ and of the number of admissible networks and as the power-law exponent is between 2 and 3 for almost all accepted degree distributions, we speculate that approximately two-thirds of all empirical, large-scale networks, which can reasonably be considered to have been drawn from some underlying infinite network, are power-law networks.

Although we have obtained positive evidence that power-law networks are not rare among larger recorded networks that have sufficient observations in the tail, for the most stringent testing scenario with $h_c = 0.015$ we tested only 500 out of the 4482 networks, whereas for the most liberal value $h_c = 0.1$ we could test slightly less than half of all networks. If the tail is not big enough, parameters estimation and testing could be misleading, generating inconclusive results about the nature of the underlying degree distribution.

## 5. Conclusion

Are power-law degree distributions rare or everywhere [42]? It is perhaps surprising that after 20 years of network science, this issue still has not been resolved and has suddenly flared up again in the scientific debate. As the question has important philosophical and conceptual consequences, it is perhaps more surprising that it has taken 20 years before careful technical reviews, such as by Voitalov *et al.* [8], have considered this question methodologically. With this current paper, we hope to have contributed to this recent methodological progress.

In this paper, we have developed a tail testing procedure, taking into account a host of issues related to testing degree distributions of a single empirical network. We have presented the behaviour of the KS statistic for the discrete degree distributions, making corrections in order to achieve an even sensitivity on the observed degrees. We have presented an alternative power-law degree distribution that can be tuned to specify the size of the deviation from the power-law, and then use it to calculate the power for the test. The degree dependency and other issues have been solved by bootstrapping the test distribution via de Solla Price growing network process. The aim of this work is to propose a rigorous approach to test with sufficient power whether sequences of dependent node degrees can be distinguished from a specific power-law distribution in the tail. What we mean by 'rigorous' is that given the definition of the modified KS test-statistic, our testing procedure is exact, i.e. with exact coverage and power, up to the

precision of the bootstrap sampling. Although a power-law is a property that has sometimes been explicitly associated with the in-degree distributions [2], our testing framework can be applied to any arbitrary degree sequence, whether in-degree, out-degree or full degree distribution, both for directed and undirected simple networks.

Our aim was to re-evaluate the conclusion from Broido & Clauset [17] by applying our testing framework to the same 4482 empirical degree distributions tested there. However, in contrast to their claim that power-law distributions are rare, we classified approximately 64% of the networks for which we have sufficient power, as power-law—and most of those as scale-free. Our conclusion is that power-law networks are not rare at all. Furthermore, we note that in this framework we just tested for power-law networks using the de Solla Price model, which is a small subclass of power-law degree networks. This suggests that an even larger number of real-world networks could be classified as power-law had we used a larger power-law class as the null. Clearly, power-law networks seem empirically ubiquitous.

Data accessibility. The network data used in this article are available at https://icon.colorado.edu.

Authors' contributions. All authors contributed to the conceptual development of the manuscript. The text was written by I.A. in collaboration with E.W. V.V. and I.S. critically commented and improved on the written manuscript.

Competing interests. We declare we have no competing interests.

Funding. All authors acknowledge the contribution of the EU COST Action COSTNET (CA15109). In particular, the COST Action funded a visit by I.S. to E.W. and I.A.

## Appendix A. Brownian bridge

For completeness, we reproduce here the standard derivation of the Brownian bridge variance for independent samples [43]. Let $X$ be a random vector of $n$ independent and identically distributed variables with marginal cdf $F$, with realization $x_1, \dots, x_n$. For a given number $x$ in the support of $F$, we define $Y(x)$ the random vector in which $Y_i(x) = \mathbb{1}_{\{X_i < x\}}$ is a Bernoulli variable. Then

$$\mathbb{E}[Y_i(x)] = F(x)$$

and

$$\mathbb{E}[Y_i(x)Y_j(x')] = \begin{cases} F(\min(x, x')) & , i = j \\ F(x)F(x') & i \neq j \end{cases}$$

The centred sample mean of $Y(x)$ is

$$\bar{Y}(x) = \frac{1}{n}\sum_{i=1}^{n} Y_i(x) - F(x).$$

Denoting $u = F(x)$ and $v = F(x')$, the covariance function of $\bar{Y}$ is

$$\text{Cov}(\bar{Y}(u), \bar{Y}(v)) = \frac{1}{n}(\min(u, v) - uv),$$

and the sample mean can be rewritten as

$$\bar{Y}(u) = \frac{1}{n}\sum_{i=1}^{n} Y_i(F^{-1}(u)) - u.$$

We define the process $y(u)$ as the limit of $\sqrt{n}\bar{Y}(u)$ when $n \to \infty$. According to the Central Limit Theorem, it is Gaussian and its covariance function is given by

$$I(u, v) = \min(u, v) - uv,$$

and thus variance

$$I(u, u) = u - u^2 = u(1 - u).$$

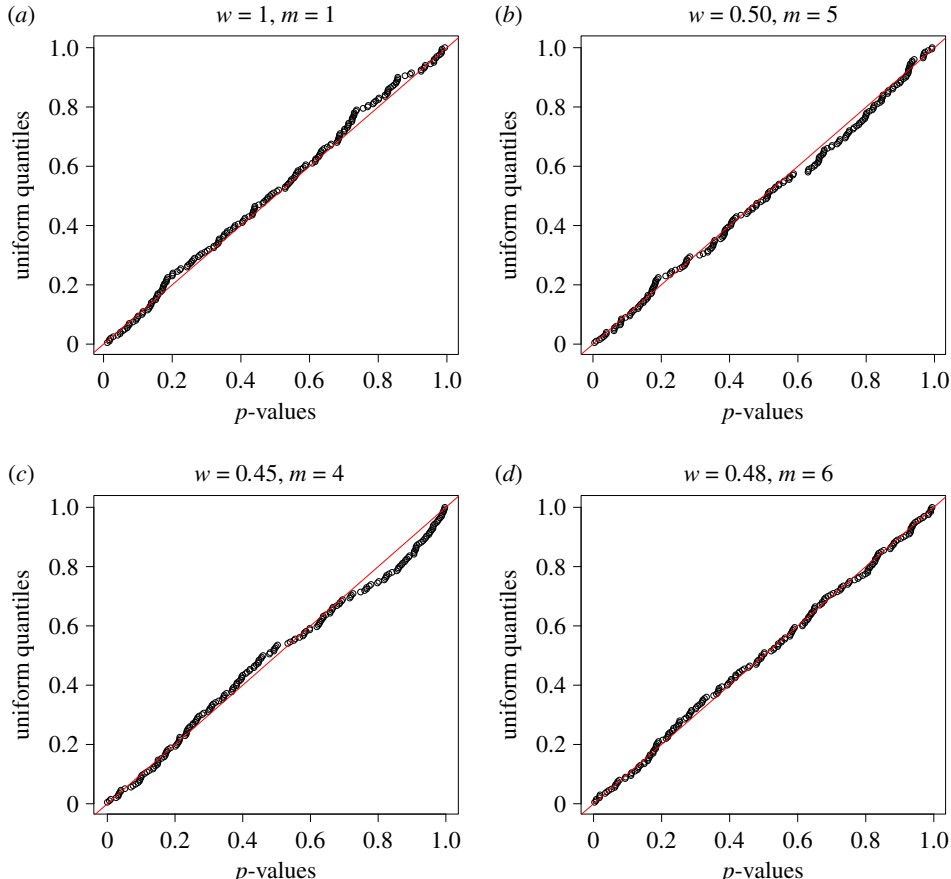

**Figure 7.** We present some qqplots of pvalues versus the quantiles of a uniform distribution, simulations performed using different parameter settings. (*a*) $w = 1$, $m = 1$, (*b*) $w = 0.50$, $m = 5$, (*c*) $w = 0.45$, $m = 4$, (*d*) $w = 0.48$, $m = 6$. (Online version in colour.)

## Appendix B. Simulation study: testing the test

A common practice when dealing with novel statistical methodologies is to run a simulation study. The aim is to check the validity of the procedure in a controlled environment. In the case of a testing procedure, this means checking the Type I Error or equivalently the uniformity of *p*-values. If the procedure is correct, we expect that the *p*-values have Uniform distribution under the null hypothesis. The simulation study is articulated as follows: for an arbitrarily fixed $(w, m)$, we simulate $B = 200$ realizations of de Solla Price degree distributions, on each of them we apply the testing procedure retrieving a *p*-value. We verify through Q-Q plot their uniformity. Finally, we repeat the simulation study for different values of $(w, m)$. Figure 7 reports some of these cases, showing that the *p*-values fit the Uniform distribution quite well, confirming the reliability of our results on the real datasets.

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
