## [Reviewer comments · Proceedings. Mathematical, Physical, and Engineering Sciences]

Review History

RSPA-2019-0742.R0 (Original submission)

Review form: Referee 1

Is the manuscript an original and important contribution to its field?

Excellent

Is the paper of sufficient general interest?

Good

Is the overall quality of the paper suitable?

Good

Can the paper be shortened without overall detriment to the main message?

Yes

Do you think some of the material would be more appropriate as an electronic appendix?

No

Do you have any ethical concerns with this paper?

No

Recommendation?

Major revision is needed (please make suggestions in comments)

Comments to the Author(s)

See report

Review form: Referee 2**Is the manuscript an original and important contribution to its field?**

Acceptable

Is the paper of sufficient general interest?

Acceptable

Is the overall quality of the paper suitable?

Acceptable

Can the paper be shortened without overall detriment to the main message?

Yes

Do you think some of the material would be more appropriate as an electronic appendix?

No

Do you have any ethical concerns with this paper?

No

Recommendation?

Major revision is needed (please make suggestions in comments)

Comments to the Author(s)

1. Overall, I think the paper is a good effort to further clarify a situation of current interest in a relevant problem (power versus not power laws in networks)

Therefore, I would feel comfortable with the idea of seeing an amended version as published in proceedings.

2. The title is misleading, since the authors consider only networks. In fact, their attention could be drawn to the fact that the potential impact of their study is broader: "fat tails" (essentially power laws) occur in many more systems, including fractals and systems exhibiting self-organized criticality.

3. The paper represents a statistics (mathematical statistics) approach to the question of scale-freeness. Some improvements of the prior analyses are presented.

4. The abstract is too much truncated. The relevance (in terms of actual probabilities) of the statement

"... classifying almost 65% of the tested networks as scale-free or power law with 80% power" has to be provided

5. The authors introduce a "modified Kolmogorov-Smirnov test", but, a little paradoxically, I do not see the test of this test (an evidence for its better suitability - in what cases?).

6. There are two questions remaining in my mind: i) Why did the authors consider "directed" graphs? ii) Is this statement indeed valid? "... finite Barabási-Albert preferential attachment model converges to a network with a power law degree distribution, other iterative

sampling schemes will not."

Decision letter (RSPA-2019-0742.R0)

04-Mar-2020

Dear Professor Wit

The Editor of Proceedings A has now received comments from referees on the above paper and would like you to revise it in accordance with their suggestions which can be found below (not including confidential reports to the Editor).

Please submit a copy of your revised paper within four weeks - if we do not hear from you within this time then it will be assumed that the paper has been withdrawn. In exceptional circumstances, extensions may be possible if agreed with the Editorial Office in advance.

Please note that it is the editorial policy of Proceedings A to offer authors one round of revision in which to address changes requested by referees. If the revisions are not considered satisfactory by the Editor, then the paper will be rejected, and not considered further for publication by the journal. In the event that the author chooses not to address a referee's comments, and no scientific justification is included in their cover letter for this omission, it is at the discretion of the Editor whether to continue considering the manuscript.

- Acknowledgements
- Funding statement

To revise your manuscript, log into <http://mc.manuscriptcentral.com/prsa> and enter your Author Centre, where you will find your manuscript title listed under "Manuscripts with Decisions." Under "Actions," click on "Create a Revision." Your manuscript number has been appended to denote a revision.

You will be unable to make your revisions on the originally submitted version of the manuscript. Instead, revise your manuscript and upload a new version through your Author Centre.

When submitting your revised manuscript, you will be able to respond to the comments made by the referee(s) and upload a file "Response to Referees" in "Section 6 - File Upload". Please use this to document how you have responded to the comments, and the adjustments you have made. In order to expedite the processing of the revised manuscript, please be as specific as possible in your response to the referee(s).

IMPORTANT: Your original files are available to you when you upload your revised manuscript. Please delete any unnecessary previous files before uploading your revised version.

When revising your paper please ensure that it remains under 28 pages long. In addition, any pages over 20 will be subject to a charge (£150 + VAT (where applicable) per page). Your paper has been ESTIMATED to be 15 pages.

Once again, thank you for submitting your manuscript to Proc. R. Soc. A and I look forward to receiving your revision. If you have any questions at all, please do not hesitate to get in touch.

Yours sincerely
 Raminder Shergill
 proceedingsa@royalsociety.org

Reviewer(s)' Comments to Author:

Referee: 1

Comments to the Author(s)
 See report

Referee: 2

Comments to the Author(s)

1. Overall, I think the paper is a good effort to further clarify a situation of current interest in a relevant problem (power versus not power laws in networks)

Therefore, I would feel comfortable with the idea of seeing an amended version as published in proceedingsa.

2. The title is misleading, since the authors consider only networks. In fact, their attention could be drawn to the fact that the potential impact of their study is broader: "fat tails" (essentially power laws) occur in many more systems, including fractals and systems exhibiting self-organized criticality.

3. The paper represent a statistics (mathematical statistics) approach to the question of scale-freeness. Some improvements of the prior analyzes are presented.

4. The abstract is too much truncated. The relevance (in terms of actual probabilities) of the statement
 "... classifying almost 65% of the tested networks as scale-free or power law with 80% power" has to be provided

5. The authors introduce a "modified Kolmogorov-Smirnov test", but, a little paradoxically, I do not see the test of this test (an evidence for its better suitability - in what cases?).

6. There are two questions having remained in my mind: i) Why did the authors consider "directed" graphs? ii) Is this statement indeed valid? "... finite Barabási-Albert preferential attachment model converges to a network with a power law degree distribution, other iterative sampling schemes will not."

Board Member:

Comments to Author(s):
 Please address the reviewers' points.

Author's Response to Decision Letter for (RSPA-2019-0742.R0)

See Appendix A.

RSPA-2019-0742.R1 (Revision)

Review form: Referee 1

Is the manuscript an original and important contribution to its field?

Excellent

Is the paper of sufficient general interest?

Good

Is the overall quality of the paper suitable?

Good

Can the paper be shortened without overall detriment to the main message?

Yes

Do you think some of the material would be more appropriate as an electronic appendix?

No

Do you have any ethical concerns with this paper?

No

Recommendation?

Major revision is needed (please make suggestions in comments)

Comments to the Author(s)

See attached report

Review form: Referee 2

Is the manuscript an original and important contribution to its field?

Excellent

Is the paper of sufficient general interest?

Good

Is the overall quality of the paper suitable?

Excellent

Can the paper be shortened without overall detriment to the main message?

Yes

Do you think some of the material would be more appropriate as an electronic appendix?

No

Do you have any ethical concerns with this paper?

No

Recommendation?

Accept as is

Comments to the Author(s)

I accept the responses

Decision letter (RSPA-2019-0742.R1)

12-May-2020

Dear Professor Wit

The Editor of Proceedings A has now received comments from referees on the above paper and would like you to revise it in accordance with their suggestions which can be found below (not including confidential reports to the Editor).

Please submit a copy of your revised paper within four weeks - if we do not hear from you within this time then it will be assumed that the paper has been withdrawn. In exceptional circumstances, extensions may be possible if agreed with the Editorial Office in advance.

Please note that it is the editorial policy of Proceedings A to offer authors one round of revision in which to address changes requested by referees. If the revisions are not considered satisfactory by the Editor, then the paper will be rejected, and not considered further for publication by the journal. In the event that the author chooses not to address a referee's comments, and no scientific justification is included in their cover letter for this omission, it is at the discretion of the Editor whether to continue considering the manuscript.

- Acknowledgements
- Funding statement

To revise your manuscript, log into <http://mc.manuscriptcentral.com/prsa> and enter your Author Centre, where you will find your manuscript title listed under "Manuscripts with Decisions." Under "Actions," click on "Create a Revision." Your manuscript number has been appended to denote a revision.

You will be unable to make your revisions on the originally submitted version of the manuscript. Instead, revise your manuscript and upload a new version through your Author Centre.

When submitting your revised manuscript, you will be able to respond to the comments made by the referee(s) and upload a file "Response to Referees" in "Section 6 - File Upload". Please use this to document how you have responded to the comments, and the adjustments you have made. In order to expedite the processing of the revised manuscript, please be as specific as possible in your response to the referee(s).

IMPORTANT: Your original files are available to you when you upload your revised manuscript. Please delete any unnecessary previous files before uploading your revised version.

When revising your paper please ensure that it remains under 28 pages long. In addition, any pages over 20 will be subject to a charge (£150 + VAT (where applicable) per page). Your paper has been ESTIMATED to be 17 pages.

Once again, thank you for submitting your manuscript to Proc. R. Soc. A and I look forward to receiving your revision. If you have any questions at all, please do not hesitate to get in touch.

Yours sincerely
Raminder Shergill
proceedingsa@royalsociety.org

on behalf of
Dr Network Science Organisers
Board Member
Proceedings A

Reviewer(s)' Comments to Author:

Referee: 2

Comments to the Author(s)
I accept the responses

Referee: 1

Comments to the Author(s)
See attached report

Board Member

Comments to Author(s):

The revision was reviewed by the original referees. As authors can see from the referee comments, there remain problems with the paper. The paper plots a careful program to unambiguously classify an observed sequence of degrees as belonging to a power-law (or scale free) distribution. The quality of their statistical approach depends on the rigor of their assumptions and definitions. Reviewer was not satisfied with the assumptions specified in the paper; hence, I am sending the paper back for a revision.

As an aside, though reviewers had not specified it, I would like authors should provide a stronger motivation about why power law/scale freeness is an important property of network degree distributions. This would benefit readers outside the discipline who may have heard of the controversy but don't know why it exists. Is it really just the presence of hubs that is important? They also exist in networks with an exponential cutoff. Is it the absence of length scale? If so, what's the big deal about it? Authors would do readers a favor of being explicit about the importance of such properties.

Author's Response to Decision Letter for (RSPA-2019-0742.R1)

See Appendix B.

RSPA-2019-0742.R2 (Revision)

Review form: Referee 1

Is the manuscript an original and important contribution to its field?

Excellent

Is the paper of sufficient general interest?

Excellent

Is the overall quality of the paper suitable?

Good

Can the paper be shortened without overall detriment to the main message?

Yes

Do you think some of the material would be more appropriate as an electronic appendix?

No

Do you have any ethical concerns with this paper?

No

Recommendation?

Accept as is

Comments to the Author(s)

Nice paper, ready to be accepted.

Decision letter (RSPA-2019-0742.R2)

25-Jun-2020

Dear Professor Wit

I am pleased to inform you that your manuscript entitled "How rare are power-law networks really?" has been accepted in its final form for publication in Proceedings A.

Our Production Office will be in contact with you in due course. You can expect to receive a proof of your article soon. Please contact the office to let us know if you are likely to be away from e-mail in the near future. If you do not notify us and comments are not received within 5 days of sending the proof, we may publish the paper as it stands.

Open access

You are invited to opt for open access, our author pays publishing model. Payment of open access fees will enable your article to be made freely available via the Royal Society website as soon as it is ready for publication. For more information about open access please visit http://royalsocietypublishing.org/site/authors/open_access.xhtml. The open access fee for this journal is £1700/\$2380/€2040 per article. VAT will be charged where applicable.

Note that if you have opted for open access then payment will be required before the article is published – payment instructions will follow shortly.

If you wish to opt for open access then please inform the editorial office (proceedingsa@royalsociety.org) as soon as possible.

Your article has been estimated as being 18 pages long. Our Production Office will inform you of the exact length at the proof stage.

Proceedings A levies charges for articles which exceed 20 printed pages. (based upon approximately 540 words or 2 figures per page). Articles exceeding this limit will incur page charges of £150 per page or part page, plus VAT (where applicable).

Under the terms of our licence to publish you may post the author generated postprint (ie. your accepted version not the final typeset version) of your manuscript at any time and this can be made freely available. Postprints can be deposited on a personal or institutional website, or a recognised server/repository. Please note however, that the reporting of postprints is subject to a media embargo, and that the status the manuscript should be made clear. Upon publication of the definitive version on the publisher's site, full details and a link should be added.

You can cite the article in advance of publication using its DOI. The DOI will take the form: 10.1098/rspa.XXXX.YYYY, where XXXX and YYYY are the last 8 digits of your manuscript number (eg. if your manuscript number is RSPA-2017-1234 the DOI would be 10.1098/rspa.2017.1234).

For tips on promoting your accepted paper see our blog post:
<https://blogs.royalsociety.org/publishing/promoting-your-latest-paper-and-tracking-your-results/>

On behalf of the Editor of Proceedings A, we look forward to your continued contributions to the Journal.

Sincerely,
Raminder Shergill
proceedingsa@royalsociety.org

Editor PRSA

Prof. dr. Ernst C. Wit
T +41 58 666 4952
wite@usi.ch

Università della Svizzera italiana
Institute of Computational Science
Via G. Buffi 13
Lugano 6904
Switzerland

www.math.rug.nl/~ernst

Appendix A

Prepared by
EW

Date
1/04/20

Our reference
O20.041

Subject
Submission to the **Network Science Special Feature** of the Proceedings of the Royal Society A

Dear Editor,

I am delighted to resubmit our manuscript entitled “*How rare are power-law networks really?*” to the Special Feature of the Proceedings of the Royal Society A. We were very pleased with the constructive comments of the referees and with the whole team we have made a big effort to respond in detail to all the questions and adjust the manuscript accordingly.

In response to referee 2 we have slightly changed the title to include the word “networks.” However, as before, this slightly ironic title does not aim to go for easy, cheap shots at other papers on this topic. Rather the opposite. It aims to show that testing for power law degree distributions is fraught with difficulties that require effort and sometimes assumptions in order to be overcome. The manuscript introduces novel methodology to be able to test for the presence or absence of a power-law property of the degree sequence, based on a single sample from a finite network.

Although there are many interesting and important network models that are *not* power laws, the power law property is one that has important philosophical and conceptual implications that is still leading to fundamental discussions in the network science community.

Kind regards, also on behalf of my co-authors, I. Artico, I. Smolyarenko and V. Vinciotti,

Ernst C. Wit
Professor of Statistics and Data Science
Institute of Computational Science (USI, Switzerland)

Response to the Referees

“How rare are power laws really?”

Referee 1

For one, I think that the abstract and the introduction should make it much clearer that the authors assume that the data comes from the de Solla Price model, which is quite an assumption and whether this is realistic is not discussed anywhere in the paper.

We have adjusted the introduction and the abstract to explain that we use a specific power law model in our manuscript. But one thing we would like to make clear is that we do *not believe* that empirical degree distribution data are De Solla Price generated, *nor is it an assumption*. We simply show that many empirical degree distribution tail data combined with the non i.i.d. distortions cannot be rejected to have a power law distribution, of which de Solla Price is a convenient example. Even though our tests have a pre-specified power (and so non-rejections are not merely the result of a lack of evidence), this power is able to detect only distributions with a small distance h from the De Solla Price model. Some of these alternatives are themselves power law distributions, but obviously some of them are not. The general idea of the paper is represented in the figure below.

In general, the discussion of power laws in networks not only centers around whether the networks have power-law degree sequences, but also whether the exponents are in between 2 and 3 or not. The infinite-variance setting is what is generally called scale-free (even though even Barabasi is inconsistent in referring to power laws or infinite-variance power laws for the term scale-free). It may be worthwhile also to consider this question, as the scale-free networks (with diverging second moment for the degrees) are rather special. It seems to me that the framework of the authors can simply be adapted to the testing problem.

We agree that the discussion of power laws often also centers on the nature of the power law exponent (and we obviously estimated them in our framework), we were somewhat worried to include too much material that might distract from the general message of the paper. However, given your encouragement, we have now incorporated in the empirical analyses in Section 4 also some information about the estimated power law exponents in the non-rejected degree distributions. In fact, for the entire range of reasonable values of h (0.015-0.1), the power law exponents are between 2 and 3, which is indeed consistent with a scale free distribution.

The discussion of preferential attachment models and the convergence of the degree distribution to a power law is rather incomplete. Do check the books by Durrett or by van der Hofstad for a more thorough discussion. Only quoting Newman and Girvan (2004) is certainly insufficient, as this bypasses all the strong results that have been obtained over the past two decades by the mathematics community.

Many thanks for the references. We have incorporated an additional discussion on the convergence of degree distributions in the manuscript in Section 3.b.

In quite a few cases, it was unclear to me what the authors are doing. For example, the bootstrap argument on pages 8-10 is unclear to me. Is this a proper bootstrap, or rather a simulation method where the de Solla Price model is simulated many times? How are the parameters in the de Solla Price model estimated? The paper writes that the MLE is computed, but does this have a closed-form expression or is a numerical optimization method used? In the overview of the testing procedure on page 10, the authors discuss an alternative hypothesis. What is this alternative?

We are aware that our procedure is somewhat involved and we apologize for not having been able to explain it adequately in the first draft of our manuscript. We have made various adjustments in our manuscript with the aim of making our estimation and testing procedure more transparent. With respect to your specific questions:

- **Bootstrap.** We perform a “proper” parametric bootstrap to estimate the distribution of the test statistic T in equation (3.3). This is done by sampling DSP degree distribution data under the MLE estimates of w and m , as described formally by Efron (1992). However, the referee is correct to

spot that in the definition of the test statistic, there is a denominator that is unknown and estimated via Monte Carlo simulation. We have adjusted the sentence to indicate that this step is a simulation estimate.

- **Parameter estimation.** Estimation is done according to MLE, whereby the parameters are estimated numerically via an iterative algorithm (David M. Gay (1990), Usage summary for selected optimization routines. Computing Science Technical Report 153, AT&T Bell Laboratories, Murray Hill.)
- **Alternative distribution.** The alternative distribution on page 10 is the least detectable distribution with an at least h_c distance from the De Solla Price power law model. This distribution is a De Solla Price distribution with a h_c jump in the tail, as shown in Figure 3. This distribution is used to calculate the power in the worst-case scenario that the true distribution is at least h_c away from the De Solla Price power law model.

The paper is centered around a topic that has attracted attention from various communities. Do make it perfectly clear how rigorous the results are, as this is a defining difference between some of the communities. Such an assessment will be quite helpful also for practitioners that may wish to use your results.

We are not quite sure how to interpret your comment. We have adjusted the conclusions in order to be clearer about the rigor of our results. The aim of the paper was to be completely rigorous in order (i) to test (ii) with *sufficient power* (iii) sequences of *dependent* node degrees (iv) whether they can be distinguished from a *specific* power law distribution (v) in the *tail*. What we mean with rigorous is that given the definition of the modified KS test-statistic, our testing procedure is *exact* (i.e. with exact coverage and power) up to the precision of the bootstrap sampling inside the procedure. We have now included several simulations (see figure under the null to show that the p-values are uniformly distributed

Our eventual conclusion that about 2/3 of a certain canon of network distributions cannot be rejected to be scale free for a consistent range of detectable deviations should be taken to mean that although subtle deviations from power law are clearly possible, a significant number of the empirical networks that we have tested – which may or, certainly, may not be a representative sample of real world networks – have strong scale free properties.

The topic of estimation in preferential attachment models has already attracted some attention. See Tiandong Wang, Sidney I. Resnick. "Degree Growth Rates and Index Estimation in a Directed Preferential Attachment Model" (2018) and Gao, F. and van der Vaart, A.W., "On the asymptotic normality of estimating the affine preferential attachment parameter" (2017). Do a good search of the literature there as well. Your results do seem to be novel, as testing has not yet been addressed. However, at least the above two papers are quite relevant to the topic under consideration.

Many thanks for the references you have provided. Indeed, the topic of parameter estimation in PA models is beginning to flourish. We have indeed been rather agnostic about the estimation aspects in our paper. The main reason is that given our testing framework required for answering our question of interest, we were more interested in the distribution of a Kolmogorov-Smirnov type test-statistic, rather

than the distribution of the PA parameter estimates themselves. However, the topic is important and we have included an additional part in Section 3.c.(iii).

We have responded to all the minor corrections directly in the paper. However, with respect to some more pertinent questions, we answer them here below:

Page 1, line -7 abstract: I have no idea what the phrase 'that achieves even sensitivity along the tail' means. Also, in general, do mention that you are considering the de Solla Price model in this paper. Now the abstract sounds too good to be true (and it is too good to be true).

Page 2, line -23: Do Voitalov et al. (2019) really perform a testing procedure? I thought they performed mainly an estimation procedure...

You are right that we cannot test for a general powerlaw distribution. In fact, that is one of the conclusions of the Voitalov et al. (2019) paper. However, our aim is still to check whether empirical networks show powerlaw behavior and the two reasons why we are taking a parametric model is that

- this circumvents the objection of Section V in Voitalov et al. (2019)
- AND we are interested in finding a kind of a lower bound of powerlaw networks. If we cannot reject that the network is De Solla Price with a guaranteed level of power, it means that there is positive evidence for the fact that this network has a powerlaw degree distribution.
- We have corrected the manuscript to make our parametric assumption clearer. The classic Kolmogorov Smirnov test is known for detecting better deviations in the midpoint of a distribution. Our modification corrects this behavior in order to detect deviations in the very end of the tail.

Page 3, line 2: Here you assume directedness of the graph. Later on, also undirected settings are discussed. Do make it clear what your setting is.

The testing procedure described in the paper is applicable to both directed and undirected graphs. As the De Solla Price model is a directed graph model, we need some adjustments to make the method applicable to undirected graphs.

Page 5, line -6: this model has rather dumb outdegrees (all constantly equal to m). Thus, when considered as a directed model, there must be much better ways to reject the model as being realistic. Do comment on this. Don't you think of the model as being an undirected network?

The referee is correct, indeed we are not testing for de Solla Price networks. Our interest is in detecting power-law networks. We consider a network as being power-law if the indegree distribution presents a power-law behavior. We are not interested in testing for de Solla Price model but just in testing if the indegree distribution of a real-world network share certain properties with the indegree distribution of a de Solla Price network.

Our testing framework is the following:

1. We observe a real-world network
2. We ask the question whether the indegree of this network is a powerlaw.
3. We make the test even more stringent asking if the indegree of this network satisfy a De Solla Price network structure.
4. We test and find that 2/3 of real networks cannot be rejected to have De Solla Price-like indegrees.
5. The final conclusion is NOT: Real networks are De Solla Price networks. The conclusion IS: Powerlaw networks are not rare.

Page 5, Section 3(a): 'These networks are not of interest to us.' Why not? Do expand on this. Note that even the Erdos-Renyi random graph can be phrased as a dynamical model (however, where edges also need to be removed). Are you only thinking about dynamic networks that have growth in them?

It is important for the statement that a finite network “comes from” a power-law network that the finite network distribution is somehow “connected” with an underlying infinite network process. There are a number of ways that this is possible. It could be that the finite network is a sample from the infinite network *or* the finite network is a finite snapshot of an infinitely growing network. Under our Null Hypothesis, we have chosen for the second option. Thus, under H_0 the degrees of the observed network are well-mimicked by a preferential attachment process.

Page 6, (3.2): This equality in distribution is not true for most networks, make this perfectly clear here.

Page 7, line 7: I do not understand this last sentence of the paragraph!

We corrected the manuscript by clarifying that this assumption is rather improper and it is used as a baseline for showing many aspects of the KS statistic. We clarified that the points (i), (ii) and (iii) must be considered deviations from this assumption.

Page 7, line -7: Why is the correlation between degrees of order $1/N$? This may be true in Erdos-Renyi random graphs, but I fail to see why this would be true for the de Solla Price model, or for preferential attachment models in general. This will depend to a large extent on which vertices are being chosen! Do make this more precise.

The degrees of empirical networks are not ordered, so the only type of correlations that is relevant here is for two randomly selected vertices. The correlations can indeed have a more general behavior. $1/N$ is the ‘strength’ at which they have a finite but not overwhelming effect on KS testing, and it is indeed shown in (Smolyarenko, Arxiv 2020) that this is the case for both Erdos-Renyi and Barabasi-Albert networks, and

we empirically validated this for the more general de Solla Price models. This is not necessarily true, however, for general preferential attachment models. For the practical test-statistic in equation (3.3), we obtained that the cumulative effect of the variance decreases due to the negative correlation between the observed degrees and the variance increases due to the observed random degrees (see below).

Page 8, Figure 2: Why does the figure for A stop? This makes little sense.

The line starts from the first value of the cumulative distribution function, for that combination of parameters the first degrees have approximately probability 0.66.

Page 8, lines 1-2: The binomial distribution in (3.2) holds for NO degrees k in most models. In general, I cannot follow this paragraph at all. Why is the variance higher than for the binomial distribution? Can you describe this in terms of something that I would be able to simulate?

We attach a “random_degrees_variance_inflation.R” script that can be used in R. It contains a replication of the Goldman and Kaplan’s example in order to show that the variance over random degrees is higher than expected compared to the Binomial case at fixed degrees. In the section we have added an explanation of why the variances are higher at the random observed degrees, rather than at fixed degrees.

Page 9, line -2: How do you compute the MLE for the parameters? Even m is somewhat non-trivial, right?

We have dedicated a few lines to parameters estimation, explaining how we deal with the discrete nature of m using a profile maximum likelihood approach: we propose a sequence of natural numbers for m , for each of them we maximize the likelihood according to w . Finally we select the combination of (w, m) with highest likelihood.

Page 10, (iii): What is the alternative distribution here? Could also an Erdos-Renyi random graph be considered as an alternative? Does the power not depend sensitively on what class of alternatives you allow for? But then the discussion of the alternatives should be much more profound, in my opinion!

We have clarified what we take for the alternative distribution in the power calculations: we consider the most difficult to detect as alternative (using the Kolmogorov-Smirnov statistic) a de Solla Price distribution (with the same parameters as estimated under the null) that is truncated at the $1-h$ quantile. Of course, we could have used an *Erdos-Renyi* as an alternative distribution too, but we use the truncated de Solla Price because it is the hardest-to-detect distribution, in order to test against the worst-case scenario. This distribution is indeed the one that minimizes the power, thus it is the hardest to reject among all the possible alternatives.

Page 10, line -1: Why have you chosen $c = 10$? Is this dependent on the data, is it an estimate, or are you eyeballing this value? Do explain.

There is a trade-off between the ability of the c -tail of a de Solla Price network to fit empirical degree distributions and the available power of our test: the higher c , the more irrelevant low degree structural effects are discarded, but also the lower the power available to test. The lower c , the higher the power, but the more likely it becomes that networks are rejected as powerlaw because of irrelevant lower degree details. We chose $c=10$ as a compromise between these two demands: for c values closer to 0 we would have an increment of rejected tests due to the inclusion of irrelevant lower degrees “in the tail”. On the other hand, for c greater than 20 we have very few networks being tested due to the lack of power in testing tails without enough points.

I miss a discussion of WHY the conclusions in this paper are so dramatically different from those by Broido and Clauset (2019). What specific part of the modeling makes the conclusion so much different? This is not clear to me at all.

We claim that both the methodology and the conclusions are completely different from Broido and Clauset (2019). The work our *Broido and Clauset (2019)* was centered on ranking power-laws into different levels of powerlaw-ness, showing that the purest level was almost never reachable. Their multitesting procedure has been criticized by Barabasi as misleading (Barabasi, 2019). Our method is a proper testing procedure (as explained in section 3) that takes into consideration all the problems with testing degree distributions (as explained in section 2). Rather than some ad hoc testing framework, we feel that our methodology can be used as a definitive way to test for the empirical evidence of the ubiquity of (some class of) powerlaw distributions.

random_degrees_variance_inflation.R

```
#replicating Goldman and Kaplan example generating from a Gaussian
n = 100
y = sort(rnorm(n, 0, 1))

cdf = pnorm(y)
emp.cdf = (1:n)/n

plot(y, cdf, ty="l")
lines(y, emp.cdf, ty="s")
plot(cdf, cdf, ty="l", main="Rescaled cdfs")
lines(cdf, emp.cdf, ty="s")
plot(cdf, sqrt(n)*(emp.cdf-cdf), ty="l", main="Brownian Bridge")
abline(h=0, col=2)

#This function takes the KS distances and rescale them according to the theoretical
variance (p(1-p)/n)
KS.weighted.distances = function(observed.cdf, real.cdf){
  sqrt(n/(real.cdf*(1-real.cdf)) ) * pmax( abs(observed.cdf - real.cdf) ,
abs(c(0,observed.cdf[-n]) - real.cdf ) )
}

B=1000
res=matrix(NA, n, B)
for(b in 1:B){
  y = sort(rnorm(n, 0, 1))
  cdf = pnorm(y)
  emp.cdf = (1:n)/n

  res[ , b] = KS.weighted.distances(emp.cdf, cdf)
}

variance = apply(res, 1, function(x) mean(x^2))

plot(cdf, variance, ty="l")
abline(h=1, col=2)
# if p*(1-p) was the proper variance of the brownian bridge then we would expect a
constant variance over the points

#generating from Exponential
B=1000
res=matrix(NA, n, B)
for(b in 1:B){
  y = sort(rexp(n, 1))
  cdf = pexp(y)
  emp.cdf = (1:n)/n

  res[ , b] = KS.weighted.distances(emp.cdf, cdf)
}

variance = apply(res, 1, function(x) mean(x^2))

plot(cdf, variance, ty="l")
abline(h=1, col=2)
```

Referee 2

Overall, I think the paper is a good effort to further clarify a situation of current interest in a relevant problem (power versus not power laws in networks). Therefore, I would feel comfortable with the idea of seeing an amended version as published in Proceedings A.

Many thanks for your kind remarks. We have made the adjustments you have suggested and respond to your comment below.

The title is misleading, since the authors consider only networks. In fact, their attention could be drawn to the fact that the potential impact of their study is broader: "fat tails" (essentially power laws) occur in many more systems, including fractals and systems exhibiting self-organized criticality.

Given that the special issue is about Networks, we did not add networks to the title. However, we can see that this is potentially misleading, especially when the paper is cited outside the context of the special issue. We have added "network" to the title. Your suggestion that the work may be able to be extended to other systems is an interesting one, but one that in the context of this special issue we are unable to pursue.

The abstract is too much truncated. The relevance (in terms of actual probabilities) of the statement "... classifying almost 65% of the tested networks as scale-free or power law with 80% power" has to be provided.

We have adjusted the abstract and hope to have made it clearer for a lay person to understand the focus and outline of the method.

The authors introduce a "modified Kolmogorov-Smirnov test", but, a little paradoxically, I do not see the test of this test (an evidence for its better suitability - in what cases?).

In the new version of the paper we have added a small simulation study about the appropriate behavior of the modified Kolmogorov-Smirnov test. Figure 7 shows that the modified test achieves the theoretical $U(0,1)$ significance.

There are two questions having remained in my mind: i) Why did the authors consider "directed" graphs? ii) Is this statement indeed valid? "... finite Barabási-Albert preferential attachment model converges to a network with a power law degree distribution, other iterative sampling schemes will not."

We answer to your two questions below:

- (i) The choice for directed or undirected networks is mainly important for the formulation of the null distribution. Given that we used the standard De Solla Price model, which is a directed

network model, our discussion has focused on directed networks. It is certainly possible to extend the testing framework to undirected graphs, as undirected versions of the De Solla Price model do exist. However, we felt that this extension would distract rather than add to the already complicated nature of our paper.

- (ii) This statement is indeed not right and should have read "... other iterative sampling schemes might not."

Editor PRSA

Prof. dr. Ernst C. Wit
T +41 58 666 4952
wite@usi.ch

Università della Svizzera italiana
Institute of Computational Science
Via G. Buffi 13
Lugano 6904
Switzerland

www.math.rug.nl/~ernst

Appendix B

Prepared by
EW

Date
8/6/20

Our reference
O20.061

Subject
Submission to the **Network Science Special Feature** of the Proceedings of the Royal Society A

Dear Editor,

I am delighted to resubmit our manuscript entitled “*How rare are power-law networks really?*” to the Special Feature of the Proceedings of the Royal Society A. We were again very pleased with the constructive comments of the referees and with the whole team we have made a big effort to respond in detail to all the questions and to adjust the manuscript accordingly. Changes are indicated in red.

Referee 2 was satisfied with our original revision. Referee 1 had a number of comments, in particular (s)he asked us to clarify how we use the De Solla Price model in our paper and secondly what parametrization we used. We are confident that this has now been resolved. Moreover, you asked us to reflect on the philosophical and conceptual importance of power-law networks. We have added this in the introduction. Again, in red.

Kind regards, also on behalf of my co-authors, I. Artico, I. Smolyarenko and V. Vinciotti,

Ernst C. Wit
Professor of Statistics and Data Science
Institute of Computational Science (USI, Switzerland)

Response to the Referees

“How rare are power laws really?”

Board member

I would like authors should provide a stronger motivation about why power law/scale freeness is an important property of network degree distributions. This would benefit readers outside the discipline who may have heard of the controversy but don't know why it exists. Is it really just the presence of hubs that is important? They also exist in networks with an exponential cutoff. Is it the absence of length scale? If so, what's the big deal about it? Authors would do readers a favor of being explicit about the importance of such properties.

Describing networks can be done in two distinct ways. Clearly from a phenomenological point of view, there is barely any difference between power law network and a large variety of other network models, such as networks with an exponential cut-off. One of the reasons is that observed networks are (almost) always finite and therefore network degrees are finite and whatever we propose for the extreme tails is – from an observational point of view – irrelevant. By analogy, if we obtain a sample from a population, whatever our model suggests being the case for the non-observed part of the population cannot be refuted logically by our data. So, if our sole purpose is fitting a degree distribution, or more ambitiously a network model, then a whole class of models will do an equally good job for the types of networks we tend to encounter in practice. However, ever since De Solla Price started to experiment with potential generative network models in the 1960s, it became clear that a small number of substantively plausible and generative principles are capable of generating network structures that correspond to empirical networks. Particularly, various forms of the preferential attachment principles have been shown to result in network structures whereby the degree sequences are generally described by ratios of gamma functions, i.e., power laws (e.g. Krapivsky and Redner, 2001). This putative universality of the power law degree distributions therefore turns the question around: although there may be many different degree distributions that correspond to empirical degree sequences, additional arguments are needed to propose ad hoc variations (such as exponential cut-offs). Power-law networks act as a natural paradigm for falsification (Popper, 1962), i.e., as a natural null hypothesis.

This does not mean that power law degree distributions do not require empirical support, rather the opposite. They definitely do require empirical validation. A practical reason why this question is important is, for example, the fact that power law networks are more susceptible to epidemics and other viral activities, such as gossip or the spread of misinformation. The universality of power law networks is thus a critically important question and resolving its empirical status would inform efforts to apply results from network theory to many scientific questions (Broido and Clauset, 2019).

Referee 1

I believe that the authors compare their network data sets to the degree distribution of a de Solla Price model, and this is how I read it. However, this is not always equally clear.

Yes, indeed, the referee is right. We compare the degrees to the degree distributions of a de Solla Price model. We have tried to clarify this throughout the text.

However, it is not the likelihood of the de Solla Price model, since there the degrees are not i.i.d. This means that the authors use an estimation procedure for their important parameters $m;w$ that is not based on the actual MLE procedure, but rather on a related MLE procedure that does not correspond to their model.

Again, the referee is correct: we do not maximize the likelihood, but a pseudolikelihood. We have corrected this in the text. Crowder (1976) showed that under general conditions this pseudolikelihood approach is consistent. Nowhere we use the particular properties of the MLE in our testing procedure, so the results are completely robust to the estimation method.

It would certainly help me if the authors were to write down explicitly how to translate the m and w parameters used here to the m and used in van der Hofstad (2013). I am unsure whether the formula for P_{inf} is incorrect.

We provide an explanation on how to relate our degree distribution with the one presented in van der Hofstad (2013). We are considering an indegree distribution where the indegree of a node is shifted by the m parameter $k^{in} = k^T - m$, where k^T is the total degree of a node used in Equation (8.3.2) in van der Hofstad (2013). It is possible to see the equivalence of the two models by switching from total degree to indegree.

The total (sum of indegree and outdegree) degree distribution for graphs, following van der Hofstad (2013) at Equation (8.3.2) is

$$p_{k^T} = (2 + \delta/m) \frac{\Gamma(k^T + \delta)}{\Gamma(k^T + 3 + \delta + \delta/m)} \frac{\Gamma(m + 2 + \delta + \delta/m)}{\Gamma(m + \delta)} \quad (0.1)$$

We reparametrize as $w = \delta + m$, obtaining

$$p_{k^T} = (1 + w/m) \frac{\Gamma(k^T + w - m)}{\Gamma(k^T + 2 + w - m + w/m)} \frac{\Gamma(1 + w + w/m)}{\Gamma(w)} \quad (0.2)$$

Now we consider the indegree distribution where $k^{in} = k^T - m$

$$p_{k^{in}} = (1 + w/m) \frac{\Gamma(k^{in} + w)}{\Gamma(k^{in} + 2 + w + w/m)} \frac{\Gamma(1 + w + w/m)}{\Gamma(w)} \quad (0.3)$$

that is the de Solla Price indegree distribution presented in the manuscript. The referee is correct in pointing out that the Barabasi-Albert model is given by $\delta = 0$ so $w = m$ and therefore

$$2 \frac{\Gamma(k^{in} + m)}{\Gamma(k^{in} + 3 + m)} \frac{\Gamma(2 + m)}{\Gamma(m)} = 2 \frac{\Gamma(k^T)}{\Gamma(k^T + 3)} \frac{\Gamma(2 + m)}{\Gamma(m)}. \quad (0.4)$$

Furthermore, to get the exact standard form of the Barabasi-Albert model, we need $w=m=1$, where d then refers to the indegree.

More detailed remarks and typos.

(1) *Page 6, line 15: Make very clear here that 'sample' does not mean i.i.d.*

We corrected the manuscript, in order to make it clear that we work with a non i.i.d. degree sequence.

(2) *Page 6, line 25: It could be that the vertex set is fixed and the edges are drawn from some distribution'*

We corrected the point in the manuscript.

(3) *Page 6, line 53: I think Broido and Clauset is meant here. Vojtalov et al. uses a very general power-law shape, so this can hardly be very restrictive.*

Yes, the referee is right. Apologies. We indeed meant Broido and Clauset (2019).

(4) *Page 7, line 17: Add 'i.e.,' before the formula.*

This has now been corrected in the manuscript.

(5) *Page 7, line 38: I still struggle with the H_0 formulated here. Indeed, H_0 only refers to the degree distribution. However, if I understand the paper correctly, actually the degree distribution of the real-world network is matched to that in the de Solla Price model, including the dependencies between the degrees (this is a major point of the paper). So, should H_0 not be that the real-word network model is the de Solla Price model?*

We agree with the referee that the proper null hypothesis is defined at the level of the network, rather than at the level of the degree distribution, as we make use of the properties of the de Solla Price network model in the calculation of the distribution of the test statistic. We have made the adjustment in the manuscript.

(6) *Page 7, line 55: I think the 'On principle" here should be replaced by 'Under the independent sample assumption'*

We made the replacement as suggested.

(7) *Page 8, line 27: 'test statistic for empirical degree distributions in evolving networks does not converge to the usual Kolmogorov limit distribution.'*

Corrected as suggested.

(8) *Page 8, (3.2): Move the assumption about 'independent sampling' to BEFORE formula (3.2), so that it is more visibly present*

We have corrected the manuscript as you suggested and clarified the sentence.

(9) *Page 9, line 55: Here you write 'The binomial expression for the scaled KS statistic, described in (3.2), holds only for fixed degrees k ', which is very confusing. You just claimed that it does NOT hold at all!*

We modified the manuscript to resolve this possible source of confusion. There are a few points to keep in mind:

- Equation (3.2) clearly does not hold for the dependent degrees case. All points (i) (ii) and (iii) on page 8 (section 3.c.ii) are intended as deviations from that baseline (independent) case. All those points are meant to be separated and non-consecutive, in the sense that one point is not based on the previous one.
- The study of empirical degree distributions shows all three factors at play. The way these three factors interact with each other is unknown and that is the reason why we generate/simulate the test statistic distribution. What we have done is analyzing them separately, reaching the conclusion that it would be complex to derive an analytic solution for the test statistic distribution.
- Those points must justify our choices in the resampling procedure. They are not meant to give any kind of information on the overall test distribution. They spot how that test distribution will not behave and what problems our testing procedure should focus on.

(10) *Page 10, line 37: Remove 'follows'*

We have corrected the text.

(11) *Page 10, line 40: Here the degrees are all of a sudden written as dv , whereas previously they were kv . I have to say that I prefer dv in the first place, as the d nicely refers to degree. Make this consistent throughout the paper.*

As suggested, we now use "d" for the degree variable consistently across all the manuscript.

(12) *Page 10, line 48: It is not quite true that \hat{w} and \hat{m} are the most likely values, since L is not the likelihood of the de Solla Price model as argued before.*

Indeed, the referee is correct. Our estimator is a type of pseudolikelihood estimator (Besag, 1977) and as Crowder (1976) showed this type of estimator is consistent, which is sufficient for our purposes.

(13) *Page 11, line 58: Again, these are not the MLEs.*

We agree. See our response to the previous comment.

(14) *Page 12, line 20: Again, do you mean d or k for the degrees?*

Corrected. See point (11).

(15) Page 15, Figure 6: The restriction to m being an integer seems rather restrictive, and this is shown rather decisively here. Certainly no real-world network will have a fixed m . Might it be an idea to allow for non-integer m by taking two subsequent values with certain probabilities? Preferential attachment models with random degrees have been investigated in [1], and these results could be used here too.

Many thanks for the reference, we have included it in the manuscript as a possible improvement. The aim of the manuscript is to find a lower bound for the rate of powerlaws. The model described in Deijfen et al. (2009) is certainly based on a more realistic assumption and we expect that it will fit the data better. As a result, the lower bound for the rate will be even higher and it will not change the final statement "powerlaws are not rare".

(16) Page 16, line 22: Do you now mean 200 realizations of the de Solla Price network model? I think so, but say this very clearly.

Yes, we clarified that point in the manuscript.

References

[1] Besag, Julian. "Efficiency of pseudolikelihood estimation for simple Gaussian fields." *Biometrika* (1977): 616-618.

[2] Crowder, Martin J. "Maximum likelihood estimation for dependent observations." *Journal of the Royal Statistical Society: Series B (Methodological)* 38, no. 1 (1976): 45-53.

[3] M. Deijfen, H. van den Esker, R. van der Hofstad, and G. Hooghiemstra. *A preferential attachment model with random initial degrees.* *Ark. Mat.*, 47(1):41{72, (2009).